# Antibacterial Screening, Biochemometric and Bioautographic Evaluation of the Non-Volatile Bioactive Components of Three Indigenous South African *Salvia* Species

**DOI:** 10.3390/antibiotics11070901

**Published:** 2022-07-06

**Authors:** Margaux Lim Ah Tock, Sandra Combrinck, Guy Kamatou, Weiyang Chen, Sandy Van Vuuren, Alvaro Viljoen

**Affiliations:** 1Department of Pharmaceutical Sciences, Faculty of Science, Tshwane University of Technology, Pretoria 0001, South Africa; limahtockmj@tut.ac.za (M.L.A.T.); combrincks@tut.ac.za (S.C.); kamatougp@tut.ac.za (G.K.); chenw@tut.ac.za (W.C.); 2Department of Pharmacy and Pharmacology, Faculty of Health Sciences, University of the Witwatersrand, 7 York Road, Johannesburg 2193, South Africa; sandy.vanvuuren@wits.ac.za; 3SAMRC Herbal Drugs Research Unit, Faculty of Science, Tshwane University of Technology, Pretoria 0001, South Africa

**Keywords:** antibacterial activity, biochemometric analysis, bioautography, high performance thin layer chromatography-mass spectrometry, *Salvia africana-lutea*, *S. lanceolata*, *S. chamelaeagnea*

## Abstract

*Salvia africana-lutea* L., *S. lanceolata* L., and *S. chamelaeagnea* L. are used in South Africa as traditional medicines to treat infections. This paper describes an in-depth investigation into their antibacterial activities to identify bioactive compounds. Methanol extracts from 81 samples were screened against seven bacterial pathogens, using the microdilution assay. Biochemometric models were constructed using data derived from minimum inhibitory concentration (MIC) and ultra-performance liquid chromatography-mass spectrometry data. Active molecules in selected extracts were tentatively identified using high-performance thin layer chromatography (HPTLC), combined with bioautography, and finally, by analysis of active zone eluates by mass spectrometry (MS) via a dedicated interface. *Salvia chamelaeagnea* displayed notable activity towards all seven pathogens, and the activity, reflected by MICs, was superior to that of the other two species, as confirmed through ANOVA. Biochemometric models highlighted potentially bioactive compounds, including rosmanol methyl ether, epiisorosmanol methyl ether and carnosic acid. Bioautography assays revealed inhibition zones against *A. baumannii*, an increasingly multidrug-resistant pathogen. Mass spectral data of the eluted zones correlated to those revealed through biochemometric analysis. The study demonstrates the application of a biochemometric approach, bioautography, and direct MS analysis as useful tools for the rapid identification of bioactive constituents in plant extracts.

## 1. Introduction

Individuals with weakened immune systems are particularly susceptible to opportunistic infections caused by common bacterial pathogens, including Gram-positive *Staphylococcus aureus*, *Bacillus cereus*, *Enterococcus faecium*, *Bacillus subtilis*, and Gram-negative *Escherichia coli*, *Acinetobacter baumannii*, and *Pseudomonas aeruginosa*. Several *Salvia* species that are indigenous to southern Africa are commonly used alone, or as an ingredient of traditional polyherbal remedies, as treatments for a variety of infectious ailments. The three species, *S. africana-lutea* L., *S. lanceolata* L., and *S. chamelaeagnea* L., which are the subject of this research, are commonly used to treat infections related to the lungs, skin, stomach, and other infectious diseases [1]. *Staphylococcus aureus*, *P. aeruginosa*, *A. baumannii*, *E. faecium*, and *B. subtilis* are related to respiratory [2] and skin ailments [3], and together with *E. coli*, are associated with urinary tract infections [4]. Stomach ailments characterised by diarrhea, are commonly caused by *B. cereus* [5], *E. coli*, or *E. faecium* [6]. If these conditions are not treated fast, dehydration and secondary infections such as septicemia and gangrene can set in, often resulting in death, especially in the case of immunocompromised patients.

The antibacterial activity of methanol:chloroform (1:1, *v*/*v*) extracts of indigenous *Salvia* species, including the three species, has been previously investigated in vitro [7,8]. Minimum inhibitory concentrations (MICs) towards *S. aureus*, *B. cereus*, *E. coli*, and *K. pneumoniae* ranged from 0.030–8.0 mg/mL, indicating excellent to weak activity [8]. The Gram-positive pathogens were more susceptible to the extracts of two of the three species, *S. africana-lutea*, and *S. chamelaeagnea*. The same research group conducted further investigations into *S. chamelaeagnea*, as it displayed the best antibacterial activity [8,9]. A methanol:chloroform (1:1, *v*/*v*) extract was fractionated using column chromatography, and the most active fraction towards the tested bacterium, *S. aureus*, was identified utilising thin layer chromatography (TLC) and a bioautographic assay. Compounds in the active fraction were identified as carnosol, oleanolic acid, ursolic acid, and 7-*O*-methylepirosmanol. The highly variable MICs recorded for each of the four pure compounds reflect differences in susceptibility of the four pathogens [8]. Their study demonstrated the potential antibacterial properties of the three species towards some pathogens and justified further studies of this aspect. The shortcomings of the previous studies [7,8] are the use of single samples from one locality and the testing of extracts towards only four pathogens associated with lung and skin conditions.

The bioactivity of a plant extract is related to the presence of secondary metabolites, which include phenolic compounds, and di- and triterpenoids [10], and is dependent on the relative ratios and levels at which they occur. Chemical differences can influence the bioactivity of any given plant extract dramatically. Substantial inter- and intraspecies chemical variation within both volatile [11] and non-volatile [12] secondary metabolites of the three species was scientifically confirmed through chromatographic analyses and chemometric modelling techniques. Such variation may influence the antibacterial activity of a single specimen, which may not be representative of the species in general. This was demonstrated in another study [13] that investigated the antibacterial activity of plants from two of the three species, *S. africana-lutea* and *S. lanceolata*, towards four pathogens (*E. coli*, *K. pneumoniae*, *B. cereus* and *S. aureus*). Extracts were prepared from wild-harvested material collected from the same plant specimen during each of the four seasons of the same year. A large degree of variation in the antibacterial activities of the extracts from different samples was reported for each of the species, and the results confirmed that plants from the same species that are exposed to different environmental factors comprise different chemical constituents that can influence the bioactivity of the extracts.

Biochemometric analysis correlates chemical spectral data with biological data [14] and is a useful tool that can be applied to pre-screen plant extracts for identifying bioactive compounds with potential commercial value, without incurring unnecessary laboratory work. By combining antibacterial data (MIC values) with chromatographic data (in this case ultra-performance liquid chromatography-mass spectrometry (UPLC-MS) data), orthogonal projections to latent structures-discriminant analysis (OPLS-DA) models can be constructed and used to predict bioactive compounds [14,15]. Thin layer chromatography-direct bioautography (TLC-DB) is a relatively inexpensive and simple technique that has been utilised for the screening of plant extracts to identify potentially bioactive compounds [16,17]. This technique utilises high performance thin layer chromatography (HPTLC) to separate molecules in plant extracts, followed by a biological detection procedure [18]. Constituents that are active toward the selected pathogens are revealed, after the application of *p*-iodonitrotetrazolium chloride (INT), by the formation of clear circular inhibition zones around the bands. Analysis of the corresponding active bands, eluted from an identical reference HPTLC plate using a TLC-interface, by direct infusion of the sample into a mass spectrometer (MS), reveals the molecular/fragment ions of the active compounds, which enables their tentative identification. The ions of the potentially bioactive compounds highlighted by biochemometric analysis can be correlated with those observed from MS analysis of the bioactive bands, in turn confirming or discounting compounds revealed by biochemometrics. This technique has been utilised for the rapid identification of bioactive compounds in a study on propolis [19].

Apart from the few articles referenced, literature on the antimicrobial activity of the solvent extracts of the three *Salvia* species was found to be scant. The work reported in the available literature raised sufficient interest to pursue further investigations. The aim of this study was therefore to conduct an in-depth investigation of the antibacterial activity of the three species, using a large sample size from various localities in South Africa, to provide a more accurate overall indication of the bioactivity of the three species, identify variation in activity between the species and within the populations, as well as chemotypes that would be desirable in terms of their bioactivity for cultivation and commercialisation purposes. The MICs determined for each sample towards selected pathogens (*S. aureus*, *B. cereus*, *E. faecium*, *B. subtilis*, *E. coli*, *P. aeruginosa* and *A. baumannii*), were combined with UPLC-MS profiling data, using chemometric methods of analysis (OPLS-DA). This biochemometric approach was followed to identify chemical compounds that may be associated with the antibacterial activity, if any, of each species against the selected pathogens. Sample extracts from each species that displayed the best antibacterial activity (lowest MICs) towards specific pathogens were further investigated by means of TLC-DB, to identify bioactive constituents from active zones. This approach was investigated as a less complicated alternative to conventional bioassay-guided fractionation that most often utilises column chromatography.

## 2. Results

### 2.1. Antibacterial Activity

The average MICs, together with the range, determined using the microdilution assay, for the methanol extracts of the three species towards seven selected pathogens, are presented in Table 1. Also listed for each species/pathogen combination, are the overall range and average of the MICs with standard deviation (SD), positive control ciprofloxacin result, the number of active/more active samples, number of inactive/less active samples, and the statistical difference testing results.

The antibacterial activities of the three species towards each pathogen were statistically compared to determine the significance of the variability between the three species using ANOVA single factor analysis. There was a significant difference (*p* < 0.05) between the activities of the three species for all seven pathogens. Comparing the three species to each other, two at a time, also indicated a significant difference in the activities of *S. africana-lutea* and *S. chamelaeagnea* for five of the pathogens (*S. aureus*, *B. cereus*, *E. faecium*, *A. baumannii* and *E. coli*). The activities of *S. africana-lutea* and *S. lanceolata* were significantly different for four of the pathogens (*B. cereus*, *B. subtilis*, *P. aeruginosa* and *E. coli*). Activities of *S. chamelaeagnea* and *S. lanceolata* were significantly different for all seven pathogens. Overall, *S. chamelaeagnea* displayed the best antibacterial activity towards the seven pathogens with average MICs 0.23–1.3 mg/mL (variance 0.007–0.64), followed by *S. africana-lutea*, with average MICs 0.52–3.0 mg/mL (variance 0.006–4.1), and lastly, *S. lanceolata* with MICs 0.46–4.2 mg/mL (variance 0.010–4.1).

Minimum inhibitory concentrations were determined for medicinally important compounds, carnosol, carnosic acid and ursolic acid, previously reported to be present in many of the extracts [12]. Activity was determined towards two selected pathogens that displayed high susceptibility across the three species toward a number of individual plant extracts, namely *B. cereus* and *A. baumannii*. Carnosol and carnosic acid displayed antimicrobial activity toward *B. cereus* and *A. baumannii*, with MICs of 0.25 mg/mL towards *B. cereus*, and 0.13 mg/mL and 0.062 mg/mL, respectively, towards *A. baumannii*. Ursolic acid displayed excellent activity against *A. baumannii* with an MIC of 0.031 mg/mL, although it was only weakly active towards *B. cereus* (MIC = 0.50 mg/mL).

### 2.2. Bioactive Compound Identification Using a Biochemometric Approach

After combining the biological data (MICs) with data from a chromatographic reference technique (UPLC-MS), screening for potential bioactive compounds within the sample extracts was achieved using a biochemometric approach. By creating a data matrix comprising both datasets and using SIMCA^®^ P + 14 software to apply multivariate analysis, an individual OPLS-DA model was constructed for each pathogen. The active/more active (Class 1) and inactive/less active (Class 2) MIC values (defined in Section 4.4) represented the Y-variables in the data matrix for each sample of the three species. These values were combined with the UPLC-MS dataset, which were selected as the X-variables [12]. Application of multivariate tools, such as OPLS-DA modelling, resulted in the identification of potential bioactive compounds for each species/pathogen combination by group-to-average comparison and through S-plots. Figure 1 is a representation of the OPLS-DA scores scatter (a) and corresponding S-plot (b) resulting from the constructed models for the pathogen/species combination of *S. chamelaeagnea*/*A. baumannii*, randomly selected for illustrative purposes. Similar plots were obtained for the other pathogen/species combinations (plots not shown). In Figure 1a, the blue dots represent the active/more active samples (Class 1), and the red dots represent the inactive/less active samples (Class 2). The blue dots on the S-plot (Figure 1a) represent the compounds associated with bioactivity. These compounds were identified from the retention time (Rt)/mass-to-charge ratio (*m*/*z*) pairs by comparison with those reported for a non-volatile metabolite variation study [12] of the same samples. The potentially bioactive compounds that were revealed for each species through the S-plots are listed in Table 2, Table 3 and Table 4, together with OPLS-DA model statistics for each species/pathogen combination.

Tentative identities of the compounds were obtained from the variation study of the non-volatile compounds of the same set of samples [12]. The compounds identified for *S. africana-lutea* extracts (Table 2) with potential bioactivity included salvianolic acid E (Rt 3.09 min), salvianolic acid B (Rt 3.88 min), dihydroxy-dimethoxyflavone derivative (Rt 8.97 min), methyl carnosate (Rt 10.15 min), epiisorosmanol methyl ether (Rt 10.50 min) and unknown compounds at Rt 9.02 and 10.71 min. Bioactive compounds linked to the activity of the *S. lanceolata* extracts (Table 3) included salvianolic acid E (Rt 3.10), rosmarinic acid (Rt 3.66 min), epiisorosmanol (Rt 7.80 min), together with several unknown compounds at Rt 6.79, 8.72, 10.36, and 10.43 min. For *S. chamelaeagnea* extracts (Table 4), epirosmanol (Rt 7.36 min), rosmanol methyl ether (Rt 9.82 min), carnosol (Rt 9.99 min), methyl carnosate (Rt 10.15 min), carnosic acid (Rt 11.39 min) and two unknown compounds at Rt 10.74 and 12.19 min were revealed as contributors to antibacterial activity.

### 2.3. Thin Layer Chromatography-Direct Bioautography

Selected plant extracts with the most promising antibacterial activity were further investigated for potentially active compounds by TLC-DB to tentatively identify active constituents and correlate to those revealed through the biochemometric approach. Therefore, the extract for each species that displayed the most promising antibacterial activity was utilised for further investigation by TLC-DB. The three extracts selected were Langebaan Sample 2 for *S. africana-lutea*, Mamre Sample 2 for *S. lanceolata* and Paarl Sample 6 for *S. chamelaeagnea*. The three pathogens most susceptible to the plant extracts, namely *B. cereus*, *A. baumannii*, and *P. aeruginosa*, were screened for inhibition by the crude extracts of the three selected samples. Only one, *A. baumannii*, was inhibited by the separated crude extracts, and subsequent TLC-DB analysis was conducted using this pathogen. Figure 2 represents the experimental set-up for TLC-DB. 

The crude extracts of the three samples inhibited the growth of *A. baumannii*, as can be seen from the resulting inhibition zones around the discs infused with the extracts (Figure 2a–c, respectively). This provides evidence for the antibacterial activity and correlates with the low MICs obtained for the three extracts towards this pathogen. Distinct inhibition zones were observed at different sites of each fingerprint in the bioautograms of the three sample extracts (Figure 2d–f), revealing bands with activity towards the pathogen. The corresponding TLC plate depicted in Figure 2g represents the HPTLC fingerprints of the three extracts before exposure to the pathogen, and indicates bands present in each fingerprint that are not clearly visible after overlay with the inoculated agar. Selected standards (Figure 2h, Track 6—rosmarinic acid, 7–carnosol, 8—carnosic acid, 9—ursolic acid, and 10—caffeic acid) were applied to the reference plate to assist in the identification of compounds at the corresponding R*f* values that may be responsible for the activity. These were analysed together with fingerprints of the extracts (Figure 2h, Track 11–13) and visualised with *p*-anisaldehyde-sulfuric acid reagent. Figure 2i indicates the TLC plate overlaid with agar inoculated with the selected pathogen, in this case, *A. baumannii*.

Comparing the bioautograms to the unexposed standards and sample fingerprints on the reference plate (Figure 2h), gave an indication of possible known compounds contributing to the antibacterial activity observed. Three compounds, carnosol, carnosic acid, and ursolic acid were identified in the fingerprint of the *S. chamelaeagnea* extract SCP6 (Figure 2h, Track 13) by band comparison, and in turn, could be correlated to the area where the inhibition zone was observed for this extract (Figure 2f). The three compounds therefore must contribute to the activity towards the pathogen. The reference standards could not be correlated to the active areas for the other two sample extracts (Figure 2d,e), indicating that none of these known compounds are responsible for the activity. The inhibition zone observed around the positive control, ciprofloxacin disc (Figure 2j), confirms that the *A. baumannii* strain tested is susceptible to conventional antibiotics and had not developed any resistance to the drug.

### 2.4. Hyphenated Thin Layer Chromatography-Mass Spectrometry Analysis of Bioactive Constituents

With the active bands revealed on the bioautograms, identification of the compounds was achieved using hyphenated MS analysis. The sites corresponding to zones of inhibition, where active constituents were extracted by use of the TLC-interface, are indicated on the bioautograms presented in Figure 3a. The corresponding TLC plate (Figure 3b) shows the actual sampling spots where extraction of Peaks 1–6 was performed using the TLC-sampler. Peaks 1–6, indicated in Figure 3a, correspond to Peaks 1–6 on the resulting chromatogram (generated by the software after infusing each sampled site) presented in Figure 3c. The active sites for *S. africana-lutea* are represented by Peaks 1 and 2, *S. lanceolata* by Peak 3 and *S. chamelaeagnea* by Peaks 4, 5 and 6. It must be noted that the observed ions are likely to be a combination of molecular ([M-H]^−^ and fragment ions (*m*/*z*) of the compounds present within the sampled active sites. The mass spectrum obtained from the active site sampled at Peak 1 is given as an example in Figure 3d, and molecular/fragment ions can be observed. Similar mass spectra were obtained for each active site sampled at Peaks 2, 3, 4, 5, and 6 (not shown). The molecular/fragment ions of components with potential activity present in the extracted bands were detected and reported in the form of a mass spectrum displaying the *m*/*z* ratios. The *m*/*z* of the ions for each peak are listed in Table 5. Molecular structures for possible compounds identified through TLC-MS are given in Figure 4.

## 3. Discussion

The microdilution assay enabled rapid, large-scale screening of extracts. The results indicated active extracts that could be earmarked for further investigation following a more focused approach. Suggested criteria for MIC values of plant extracts available from recently published literature [20] were considered as a guideline for activity. According to the author, for the purposes of ethnopharmacological research studies, MICs higher than 0.32 mg/mL reflect extracts that may still be effective, although 1.0 mg/mL was accepted as the upper limit of activity for this study. Activities were further described [20] as weak (>0.32 mg/mL), average (0.16–0.32 mg/mL), good (0.081–0.16 mg/mL), very good (0.041–0.080 mg/mL), excellent (0.021–0.040 mg/mL) or outstanding (<0.02 mg/mL), and these criteria were applied to this study.

For *S. africana-lutea* and *S. lanceolata*, most of the samples inhibited the growth of *B. cereus*, *A. baumannii*, *P. aeruginosa* and *B. subtilis*, with weak, average, or good activities (MIC ≤ 1.0 mg/mL). *Salvia chamelaeagnea* extracts displayed the highest degree of activity against the seven selected pathogens. Most extracts displayed weak to excellent activity, with MICs ranging from 0.038 to 1.0 mg/mL. All samples were active against *B. cereus*, *A. baumannii* and *P. aeruginosa* and most extracts were active towards the remaining four pathogens, *S. aureus*, *E. faecium*, *B. subtilis*, and *E. coli*. One sample from the Paarl population (Sample 6) displayed excellent activity towards *B. cereus* (MIC = 0.038 mg/mL), with good activity observed for 12 other samples with an MIC of 0.13 mg/mL. The same Paarl sample displayed good activity (MIC = 0.13 mg/mL) towards *A. baumannii*, which is notable, as this pathogen is known to be highly resistant towards conventional antibacterial agents [21]. Promising activity was exhibited by extracts for all three species towards *A. baumannii*. Variation in the MICs for samples from the same species and within populations was evident (Table 1) and confirms that differences in the chemical profiles of the plant extracts affect their bioactivity.

The extracts of *S. chamelaeagnea* contain high concentrations of carnosol (Not detected (ND)–15.1 mg/g dry weight (DW)) and carnosic acid (0.94–14.0 mg/g), as previously determined using ultra-performance liquid chromatography-photodiode array (UPLC-PDA) detection analysis [12]. The two compounds were found to be absent or not detected in the samples of *S. africana-lutea* and *S. lanceolata*. It is possible that carnosol and carnosic acid contribute towards the highly promising antimicrobial activity of several *S. chamelaeagnea* extracts. These compounds are present at substantial levels in sample extracts of *Rosmarinus officinalis*, which has widely reported antibacterial activity [22,23]. The activities of the carnosol and carnosic acid reference standards, as determined in this study, were lower than expected towards *B. cereus* and *A. baumannii*, with MICs of 0.25 mg/mL for both towards *B. cereus* and 0.13 mg/mL and 0.062 mg/mL, respectively, towards *A. baumannii*. The activity of carnosol towards *B. cereus* (MIC = 0.25 mg/mL) does not correlate with the higher activity (MIC = 0.02 mg/mL) previously reported [8] towards the same strain pathogen (ATCC 11778). No other reports on MICs for the two compounds towards the two pathogens could be found, apart from one study that reported a MIC of 0.078 mg/mL for carnosic acid towards *B. cereus* [24]. This value indicates a higher activity than that determined in this study (MIC = 0.25 mg/mL). It is therefore highly plausible that the two compounds, carnosol and carnosic acid, together with other compounds in the extracts, may contribute to the good antimicrobial activity of the extracts in which they are present towards *B. cereus* and *A. baumannii*, by acting in an additive or synergistic manner.

Ursolic acid may also contribute to a large extent to the antibacterial activity of the *S. chamelaeagnea* extracts, as well as that of the *S. africana-lutea* and *S. lanceolata* extracts, against the two pathogens. In this study, ursolic acid was found to be present at high concentrations in all samples representing the three species, with concentrations ranging from 7.40–38.2 mg/g [12]. The compound has been indicated in several studies to be responsible for antibacterial activity, on its own, in combination with other compounds, or as a main constituent of plant extracts. A study investigating the activity of constituents of *R. officinalis* extracts towards methicillin-resistant *S. aureus* (MRSA) reported that fractions containing ursolic acid resulted in total inhibition [25]. For ursolic acid on its own, MICs of 0.032 mg/mL, 0.064 mg/mL and 0.51 mg/mL, respectively, were reported towards *S. aureus*, *E. coli*, and *P. aeruginosa* [26]. In the present study, the compound displayed excellent activity against *A. baumannii* with a MIC of 0.031 mg/mL, although it was only weakly active towards *B. cereus* (MIC = 0.50 mg/mL). In a separate study [27], aimed at investigating the biofilm inhibitory activity of triterpenoids including ursolic acid, towards *A. baumannii*, an additive interaction of ursolic acid and ciprofloxacin in combination was reported, with a resulting MIC of 0.075 µg/mL for the combined compounds [27], which reflected better activity than the MIC of 0.16 µg/mL recorded for ciprofloxacin on its own. Other diterpene derivatives, rosmanol and 12-*O*-methyl carnosic acid, reported to have antimicrobial activity in *Salvia officinalis* extracts [28], are present in the extracts of the three species [12] tested in this study, and may contribute to the activity. It is highly likely that these compounds may act additively, or display synergistic interactions [29] with other constituents in the extracts.

In a separate study investigating the antimicrobial activity, the three species alongside 13 other South African *Salvia* species, were tested towards three of the seven pathogens used in the current study, namely *B. cereus*, *E. coli* and *S. aureus* [8]. *Salvia africana-lutea* was reported to be only weakly active towards *B. cereus* (MIC = 0.75 mg/mL) [8], which agrees with the results from the current study (Table 1). All three species were inactive toward *E. coli*, which is consistent with our findings that extracts were mainly inactive toward *E. coli*. They also reported that *S. africana-lutea* displayed weak activity towards *S. aureus* (MIC = 0.75 mg/mL). However, in the current study, most of the sample extracts were inactive toward this pathogen.

Kamatou and co-workers [8] reported *S. lanceolata* to be inactive towards the three pathogens, *B. cereus*, *E. coli* and *S. aureus*. In the current study, this was also the case for most extracts, with poor activity observed towards *S. aureus* and *E. coli*. However, the majority of samples were active towards *B. cereus* and for only a few samples originating mostly from the same populations, inactivity towards *B. cereus* was observed, affirming the insufficiency of a single sample result. In the same study [8], *S. chamelaeagnea* was reported to display very good activity towards *S. aureus* (MIC = 0.06 mg/mL) and *B. cereus* (MIC = 0.03 mg/mL), with weak activity (MIC = 1.0 mg/mL) towards *E. coli*. The results from the current study compare favourably with those reported by Kamatou and co-workers [8], since *S. aureus* was inhibited by a number of samples (weak and average activity), all samples were active toward *B. cereus* (weak to very good activity), and most samples tested displayed weak or average activity towards *E. coli* (Table 1).

*Salvia officinalis*, well known for its medicinal properties and culinary use, was reported to be inactive towards *S. aureus* (MIC = 5.0 mg/mL) and *E. coli* (MIC = 2.5 mg/mL), with weak activity against *B. cereus* (MIC = 0.63 mg/mL) [30]. This correlated to an extent with the activities observed for the three species towards the same three pathogens in the current study, with most extracts inactive towards *S. aureus* and *E. coli*, and mostly weak activity of *S. africana-lutea* and *S. lanceolata*. Another study reported *S. officinalis* to be only weakly active toward *S. aureus* (MIC = 0.94 mg/mL), but inactive towards *E. coli* and *P. aeruginosa* [31], corresponding to the inactivity observed in our study for *S. africana-lutea* and *S. lanceolata* towards *S. aureus* and *E. coli*. Again, there are differences in the reported results and those of the current study in relation to *P. aeruginosa*, with all samples of the three species screened displaying some level of activity.

In summary, the methanol plant extracts of *S. africana-lutea*, *S. lanceolata*, and *S. chamelaeagnea* exhibited potential as antibacterial agents towards *B. cereus*, *A. baumannii*, *P. aeruginosa*, and *B. subtilis*. *Salvia chamelaeagnea* was active against all seven pathogens, with very good and excellent activity reported for a few sample extracts against *B. cereus* and *A. baumannii*. The activity of the three species, and especially the activity of *S. chamelaeagnea*, against *A. baumannii*, is noteworthy. The pathogen has been reported to be persistent due to its resistance to most conventional antibiotic agents [32,33,34]. Statistically, there was a significant difference between the activities of the three species for the seven pathogens. This information supported the need for further investigation to reveal the active constituents in the extracts with the best antibacterial activities, initially using biochemometric analysis as a screening technique, followed by the more focused TLC-DB technique.

Biochemometric modelling was applied in this study to correlate the chemical components present in the highly complex extracts to the biological activity data, to assist in the identification of compounds possibly responsible for the bioactivity. Several of the compounds identified using this technique (Table 2, Table 3 and Table 4) have been reported to display antibacterial activity and are present in plant extracts with reported activity towards bacteria, therefore possibly contributing to the activity observed in this study. Salvianolic acid B, present in *Salvia miltiorrhizae*, tested using the microtiter plate assay, was reported to display potent antibacterial activity towards *Neisseria meningitidis*, a life-threatening human pathogen in some African countries [35]. Carnosol and carnosic acid, together with ursolic acid, which was not highlighted as a bioactive compound, have been reported to be responsible for the antibacterial activity of extracts of sage or *S. officinalis* toward MRSA [36] when tested using the microdilution method [37,38]. *Salvia officinalis* leaves are rich in methyl carnosate, a diterpene compound with good antibacterial activity towards *B. cereus* [39].

Methanolic extracts of *Rosmarinus officinalis* leaves, rich in rosmarinic acid, carnosol, and carnosic acid, were tested for antibacterial activity using the disc diffusion technique [23], and found to inhibit *S. aureus* and *Listeria monocytogenes*. Samples containing a higher concentration of carnosol relative to that of carnosic acid were reported to be more active. In the current study, this was the case with two *S. chamelaeagnea* samples, one from Paarl (MIC = 0.38 mg/mL) and one from Du Toitskloof (MIC = 0.75 mg/mL), towards *S. aureus*. Both samples displayed better antibacterial activity compared to the other samples for the respective populations. Overall, *S. chamelaeagnea* displayed the best activity towards the seven pathogens, in particular *B. cereus*, *A. baumannii*, and *P. aeruginosa*, possibly due to the presence of carnosol and carnosic acid. The two compounds were present at the required concentration ratios (carnosol > carnosic acid) in many of the active samples (Concentrations for individual samples given in the Supplementary material of [12]. For *S. chamelaeagnea* samples that were inactive towards *S. aureus*, carnosol was either present at a lower concentration than carnosic acid (samples from Paarl, Du Toitskloof, and Elandsberg) or the concentrations of the two compounds were almost equal (Elandsberg samples).

Many of the compounds identified for the three species have been reported in the literature to display antibacterial activity, or to be present in plant extracts with reported activity [36,39,40], supporting the strength of the biochemometric technique in highlighting compounds with antibacterial potential. Although biochemometric analysis can be used to identify compounds for preliminary screening, confirmation of their activity and identification is needed using a biological assay and a positive identification technique, such as TLC-DB and MS analysis, respectively. The application of biochemometric analysis was regarded in this study as a reliable indicator of bioactive compounds that could guide the targeted isolation of compounds with potential activity. It must be noted that ursolic acid was not highlighted as a bioactive compound from biochemometric analysis. Nevertheless, it has been reported to display antibacterial activity on its own [41] and to contribute to the activity of *S. officinalis* extracts [36]. The compound was present in all samples at relatively high concentrations and its contribution to the antibacterial activity of extracts for the three species should not be disregarded.

The TLC-DB technique has been successfully applied in various studies to indicate active constituents in bioautograms of other plant extracts that include *S. officinalis*, *Thymus vulgaris*, *Mentha piperita* extracts [28,42] and *Helichrysum pedunculatum* extracts [43]. Leaves from various South African tree species [44] and bark extracts from the *Morus alba* tree, a popular Traditional Chinese Medicine (TCM) [45] have also been assayed to reveal bioactive constituents. Antimicrobial compounds present in propolis, a resinous substance produced by honey bees, were revealed using TLC-DB, indicating the functionality of the technique with a variety of sample matrices [19]. In this study, no zones of inhibition were formed around the disks saturated with the crude sample extracts of the three species towards *B. cereus* and *P. aeruginosa*, and therefore no further investigation was conducted for these two pathogens. It is possible that the physico-chemical properties of the active constituents within the crude extracts are not optimal to promote sufficient diffusion into the water-based agar layer, resulting in no inhibition of the pathogen. 

From the mass spectra resulting from TLC-MS for *S. africana-lutea*, the observed ions at *m*/*z* 283, 315, 329, and 359 could be correlated to the molecular/fragment ions of rosmaridiphenol (Rt 9.76 min), rosmanol methyl ether (Rt 9.81 min), epiisorosmanol methyl ether (Rt 10.50 min), and seven unknown compounds at Rt 8.71, 9.07, 10.29, 10.71, 11.50, 12.19, 12.71 min (Table 5). Compounds at Rt 9.07 and 10.71 min corresponded to the active compounds identified via biochemometric analysis. A prominent ion at *m*/*z* 401 could not be explained. The *m*/*z* of the ions observed at 345, 315, and 359 for *S. lanceolata* correlated to epiisorosmanol (Rt 7.83 min), rosmaridiphenol (Rt 9.76 min), epiisorosmanol methyl ether (Rt 10.50 min), and an unknown compound at Rt 6.52 min (Table 5). Epiisorosmanol was indicated as an active compound by biochemometric analysis. The presence of prominent ions at *m*/*z* 347 and 401 could not be associated with specific compounds. However, *m*/*z* 401 was also observed in the spectra of *S. africana-lutea*. A few other less prominent ions were observed that could not be correlated to any compounds. Mass spectra of *S. chamelaeagnea* highlighted ions at *m*/*z* 373, 315, 285, 329, 331, 345, 287, and 455 that correlated to six compounds, namely methyl rosmarinate (Rt 4.57 min), rosmaridephenol (Rt 9.76 min), carnosol (Rt 9.98 min), methyl carnosate (Rt 10.11 min), carnosic acid (Rt 11.39 min) and ursolic acid (Rt 13.80 min) (Table 5). Carnosol and carnosic acid corresponded to active compounds indicated by biochemometric analysis and both compounds, together with ursolic acid were confirmed to be present in the extract by means of reference standards, as illustrated in Figure 2h. A prominent ion was observed at *m*/*z* 343 that could not be explained. This ion was absent from the mass spectra of the other two species. It must be noted that the compounds tentatively identified from biochemometric analysis and TLC-DB that displayed antibacterial activity may not be active on their own as such, but the activity may be due to their combination with other compounds in the plant extract. Therefore, these compounds need to be studied on their own and in combination with each other to determine their true activity or the contribution they make.

Screening and identification of plant extracts for potential antibacterial constituents using biochemometric analysis combined with a TLC-DB approach in tandem with MS identification, is a promising alternative with several advantages that make it worth considering. It is generally faster, simpler, less expensive, specific, sensitive, and more environmentally friendly [28,46,47] than routinely used bioassay-guided fractionation, which is labour intensive, uses large quantities of plant material and solvents, and primarily focuses on major compounds that may not present bioactivity on their own [14,46]. Many samples can be screened simultaneously, and infusion of eluates into the MS is a direct approach that delivers identifications almost instantaneously, by providing information on compound ions [19]. Nevertheless, there are some limitations to this approach. With biochemometric analysis, the large number of variables from the reference dataset may complicate data visualisation and interpretation from the S-plot, resulting in false-positive identifications [14]. It is likely that some bioactive compounds are not identified, and were marked a false negative [48]. It may be due to specific peaks affected by poor separation affecting identification, as is thought to have been the case with ursolic acid, a highly bioactive and main compound in all extracts used in this study. Thin layer chromatography-bioautography may be less sensitive to the more lipophilic compounds that are not compatible with the water-based agar and thereby fail to produce inhibition zones. In addition, bands that display inhibition zones may represent a mixture of active and inactive constituents, since the resolution achieved on an HPTLC plate is not at the same level as that of more sophisticated techniques such as UPLC [47].

## 4. Materials and Methods

### 4.1. Plant Material Collection, Preparation, and Extraction

The collection of the aerial parts from various localities in South Africa, species verification, and preparation of the samples of *S. africana-lutea* (*n* = 30), *S. lanceolata* (*n* = 25), and *S. chamelaeagnea* (*n* = 26), as well as voucher specimen details, are described in two previously published articles [11,12]. Extraction of the dried aerial plant parts was performed using methanol [12]. However, after the combined supernatants of each plant extract were dried and weighed, 32 mg/mL solutions were prepared in acetone for MIC determinations. Three selected plant extracts were also prepared in acetone at 10 mg/mL for TLC-DB.

### 4.2. Reagents, Chemicals, and Pathogens

Tryptone Soya broth (TSB) (Oxoid; product code CM0129) and Tryptone Soya agar (TSA) (Oxoid; product code CM0131) were obtained from Quantum Biotechnologies (Pty) Ltd. (Johannesburg, South Africa) and prepared according to the manufacturer’s instructions. *p*-Iodonitrotetrazolium chloride (INT, Sigma-Aldrich, St Louis, MI, USA) was purchased from Merck (Germiston, South Africa). Ciprofloxacin (Sigma-Aldrich product number 17850, ≥98.0%, HPLC) was obtained from Merck (Pty) Ltd. (Modderfontein, South Africa). Seven bacterial strains, namely *S. aureus* (ATCC 25923), *A. baumannii* (ATCC 19606), *B. cereus* (ATCC 11778), *E. coli* (ATCC 8739), *E. faecium* (ATCC 27270), *P. aeruginosa* (ATCC 27853) and *B. subtilis* (ATCC 6051), were purchased from Davies Diagnostics (Pty) Ltd. (Randburg, South Africa). All solvents (AR grade), sulfuric and formic acids (AAR grade) were obtained from Merck (Pty) Ltd. (Modderfontein, South Africa). Ciprofloxacin discs (5 µg, Oxoid; product code CT0425B), were purchased from Thermo Fisher Scientific, Randburg, South Africa. Reference standards of individual compounds rosmarinic acid (≥98%), caffeic acid (≥98%), carnosol (100%), carnosic acid (97.6%) and ursolic acid (≥90%) were purchased from Merck (Pty) Ltd. (Modderfontein, South Africa). High-performance thin layer chromatography was performed using 20 × 10 cm HPTLC silica gel 60 F_254_ chromatographic plates (Merck KGaA, Darmstadt, Germany).

### 4.3. Antibacterial Activity Determination Using the Microdilution Assay

The MIC for each sample extract was determined towards each of the seven pathogens using the in vitro microdilution assay [49] with modifications [50]. The selected pathogens were inoculated in TSB at 37 °C for 24 h in an incubator (EcoTherm, Hartkirchen, Austria). The resulting cultures were diluted with fresh TSB to yield a concentration of approximately 1 × 10^6^ colony forming units (CFU)/mL inoculum, which was estimated by visual comparison of turbidity to a McFarland standard (0.5) solution. Into each well of a sterile 96-well microtiter plate, 100 µL of sterile TSB broth was plated. This was followed by adding 100 µL of 32 mg/mL plant extract solution in acetone and serial dilution followed. All determinations were conducted in duplicate. The final concentrations of the plant extract in the wells from rows A to H ranged from 8.0 to 0.075 mg/mL, halving with each dilution. After sealing the plate with clear tape, it was placed overnight in an incubator at 37 °C. A 40 µL volume of INT in water solution (0.04% *w*/*v*) was subsequently plated into each well. 

The plates were observed for colour change after 2–6 h, the time depending on the pathogen as indicated by the growth controls, due to the interaction of the INT solution with the bacterial organisms. The growth controls were prepared for each pathogen by mixing 100 µL TSB with the pathogen inoculum. Sterile water was used as the negative control by adding 100 µL to a well before inoculation. Positive controls were prepared by adding 100 µL of a ciprofloxacin solution (0.01 mg/mL in water) to a well and a culture control sample to confirm bacterial growth. The MIC value was determined as the lowest concentrations in each column of the microplate where no pink/purple colour was visible. The MICs for the pure compounds, carnosol, carnosic acid and ursolic acid, dissolved in methanol and further diluted in acetone, were determined using the same method, towards *B. cereus* and *A. baumannii*, two of the pathogens that were inhibited at promising MIC levels by the sample extracts. The antibacterial activities of each of the three species towards the seven pathogens were statistically compared by ANOVA single factor analysis. Significant differences between the three species were confirmed where *p* < 0.05.

### 4.4. Identification of Bioactive Compounds Using Biochemometric Analysis

Potentially bioactive compounds were identified using an untargeted OPLS-DA approach [14]. The averages of the duplicate MICs determined for the samples from each species for a specific pathogen, were combined with the corresponding UPLC-MS dataset that was utilised for a previous variation study [12]. Using SIMCA^®^ P+ 14 software (Umetrics, Umea, Sweden), an individual OPLS-DA model was created for each pathogen. Class allocation was conducted according to activity. Samples with corresponding MIC values ≤ 1.0 mg/mL were categorised as active (Class 1), while those with MIC values > 1.0 mg/mL, as inactive (Class 2).

For a few pathogens, all sample extracts were active (MIC values ≤ 1.0 mg/mL), and the resulting MIC values were therefore classed as more active and less active, this resulted in two groups to enable the construction of a discriminant analysis (OPLS-DA) model. Bioactive compounds were thus identified for the more active samples only, and these varied from pathogen to pathogen. *Pseudomonas aeruginosa* was inhibited by all the samples representing the three species and therefore, for *S. africana-lutea* and *S. chamelaeagnea*, samples with MIC values ≤ 0.50 mg/mL were classed as more active (Class 1) and MIC values > 0.50 mg/mL as less active (Class 2). However, for *S. lanceolata*, most of the MICs for the extracts were below 0.38 mg/mL and therefore MIC values ≤ 0.38 mg/mL were classified as more active (Class 1), and the remaining MIC values > 0.38 mg/mL as less active (Class 2). The same approach was followed for the other pathogens that were susceptible to all the samples. The specific values selected are indicated in Table 6.

Loadings S-plots revealed unique variables for each species/pathogen combination that were separated from the rest of the data matrix. These variables were confirmed by group-to-average comparison and by the contribution plot. Retention time/mass-to-charge ratio pairs (Rt/*m*/*z*) were revealed and compared to those previously identified in a separate study from mass spectral data [12].

### 4.5. Thin Layer Chromatography-Direct Bioautography

The chromatographic fingerprints of selected extracts were prepared by HPTLC to separate the crude extract into its components and apply biological detection of compounds with antimicrobial activity using TLC-DB [19]. The sample extracts (20 µL) were applied (10 mg/mL) to the HPTLC plates as 6 mm bands and developed to an 80 mm endpoint using a semi-automated HPTLC system (CAMAG, Muttenz, Switzerland). The plates loaded with extract were developed with the optimised mobile phase (ethyl acetate:toluene:formic acid, 30:60:10). One of the smaller developed plates was kept aside to mark the identified active sites or zones for further HPTLC-MS analysis, and another was sprayed with *p*-anisaldehyde for visualising compounds, and this served as a reference plate for identification of compounds.

A petri dish filled with 15 mL of TSA to form a thin layer, was left to solidify. After the developed plate was sterilised under UV light for an hour in a sterile laminar flow cabinet, it was placed face-up onto the surface of the solid TSA layer. The plate was then covered with a layer of pathogen-inoculated TSA. Three pathogens, that were inhibited at the most promising MIC levels by the extracts, were used for TLC-DB, namely *B. cereus*, *A. baumannii* and *P. aeruginosa*. Sterile paper discs (prepared in-house by punching filter paper and subsequent autoclaving), each infused with 40 µL of the 10 mg/mL solution of a crude sample extract corresponding to the extract applied to the plate, were placed on the surface of the solidified inoculated TSA layer to evaluate inhibition by the whole extract. A ciprofloxacin disc (5 µg) was also placed on the surface of the inoculated TSA and used as a positive control. This set-up was placed in the incubator overnight at 37 °C, where after INT solution was gently poured onto the surface of the inoculated TSA. The pink/purple colour formation was observed where bacterial growth took place, while the cream white areas indicated zones of inhibition around active compounds on the bioautograms [28], crude extract discs, and positive control. 

### 4.6. Identification of Bioactive Compounds Using Hyphenated Thin Layer Chromatography-Mass Spectrometry

The HPTLC fingerprints of the sample extracts were prepared as described in Section 4.5. The resulting plate is a separate plate, but identical to the plate used for TLC-DB, as described in Section 4.5. The target zones to be extracted were marked according to the corresponding retardation factor (R*f*) of the active zones observed after TLC-DB. The zones with possible bioactivity were extracted one-by-one from the prepared plate using the semi-automated CAMAG TLC Interface (CAMAG Laboratory, Muttenz, Switzerland). Acetonitrile was used as eluent at a flow rate of 0.1 mL/min (run time: 1 min). The eluent was then analysed in negative (ESI) mode following direct inlet into the quadrupole-Time-of-Flight-mass spectrometer (QToF- MS) (Waters Xevo^®^ G_2_ QToF, Waters, Milford, MA, USA) using MS settings as previously published [12]. The mass spectrum obtained revealed the *m*/*z* of ions that were compared and correlated to those reported in the previous study, enabling tentative identification of the compounds. The identified compounds were also correlated with those revealed through biochemometric analysis.

## 5. Conclusions

Antimicrobial screening indicated promising bioactivity for the majority of the extracts towards *B. cereus*, *A. baumannii*, *P. aeruginosa*, and *B. subtilis*, confirming the potential of these species as antibacterial agents, and substantiating their use as traditional remedies for infections associated with the pathogens. The large sample size indicated different antibacterial activities for samples from the same population, confirming the importance of considering chemical variation by incorporating many samples into the research design. *Salvia chamelaeagnea* was active towards all seven of the selected pathogens, displaying encouraging activity towards *B. cereus* and *A. baumannii*. Biochemometric analysis revealed compounds with potential bioactivity. Sample extracts with the best antibacterial activity identified through MIC screening were used for TLC-DB. Bioactive zones were highlighted by TLC-DB for extracts of the three species towards *A. baumannii*, a refractory pathogen known to be resistant to many antibacterial drugs. The technique demonstrated in this study is relatively simple for revealing potential bioactive constituents in plant extracts. Phytoconstituents present in the active zones were extracted and analysed by direct MS, which enabled the identification of potential bioactive compounds, by using data generated during the previous chemical variation study as a guide. Some of the identified compounds correlated to those predicted through biochemometric analysis. This serves as verification that a biochemometric approach can be used for predicting compounds with a high probability of good antibacterial activity. The use, in sequence, of these assays and tools, namely chromatographic analysis, the microdilution assay, biochemometric data analysis, bioautography, and HPTLC-MS, is a relatively simple, rapid, functional, and environmentally friendly approach for identifying bioactive constituents present in medicinal plants. Valuable insights into the antibacterial activity of *S. africana-lutea*, *S. lanceolata*, and *S. chamelaeagnea* were gained in this study that can be implemented during the selection of chemotypes for commercial development.

## Figures and Tables

**Figure 1 antibiotics-11-00901-f001:**
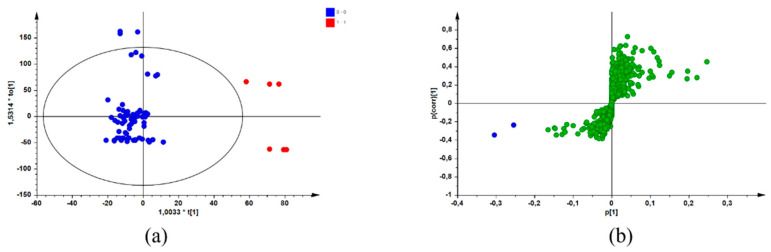
OPLS-DA scores scatter plot (**a**) with the blue dots representing active/more active samples and the red dots representing inactive/less active samples, and the corresponding S-plot (**b**) with the blue dots representing potential compounds associated with bioactivity (Model constructed from *S. chamelaeagnea*/*A. baumannii* UPLC-MS/MIC combined datasets) and the green dots representing other compounds not associated with bioactivity.

**Figure 2 antibiotics-11-00901-f002:**
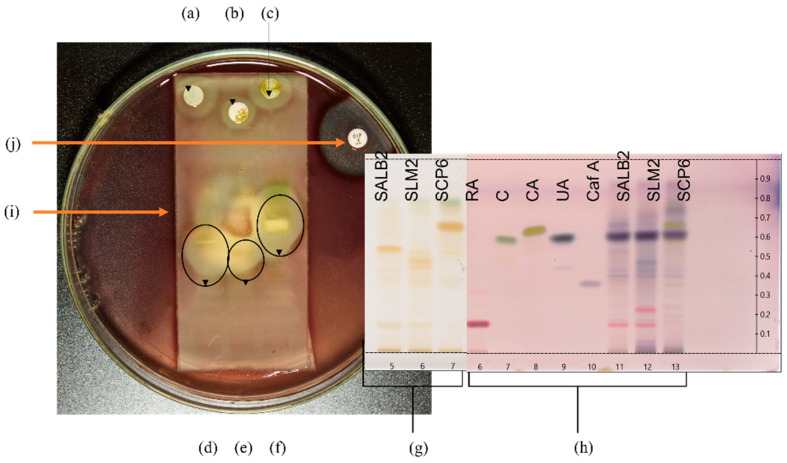
Set-up for TLC-DB with HPTLC fingerprints and bioautograms. (**a**–**c**) represent sterile discs infused with crude sample extracts of *S. africana-lutea*, *S. lanceolata* and *S. chamelaeagnea*, respectively, with inhibition zones observed around the discs indicating activity of the whole extract. HPTLC fingerprints, (**d**–**f**) of the sample extracts, *S. africana-lutea*, *S. lanceolata* and *S. chamelaeagnea*, respectively, with inhibition zones around active constituents indicated by the black ovals. (**g**) represents HPTLC fingerprints in Tracks 5, 6 and 7 for each sample extract, SALB2, SLM2 and SCP6, on the TLC plate, as viewed under white light at 366 nm, before TLC-DB. TLC plate (**h**) is the reference plate and represents authentic standards and sample extracts visualised using *p*-anisaldehyde. Tracks 6, 7, 8, 9 and 10 represent rosmarinic acid (Rf = 0.16), carnosol (Rf = 0.59), carnosic acid (Rf = 0.66), ursolic acid (Rf = 0.64) and caffeic acid (Rf = 0.35), respectively, Tracks 11, 12 and 13 represent each sample extract for identification of constituents. (**i**) represents the TLC plate overlaid with agar inoculated with the selected pathogen, in this case *A. baumannii*. (**j**) is the positive control (ciprofloxacin) disc with inhibition zone observed around the disc.

**Figure 3 antibiotics-11-00901-f003:**
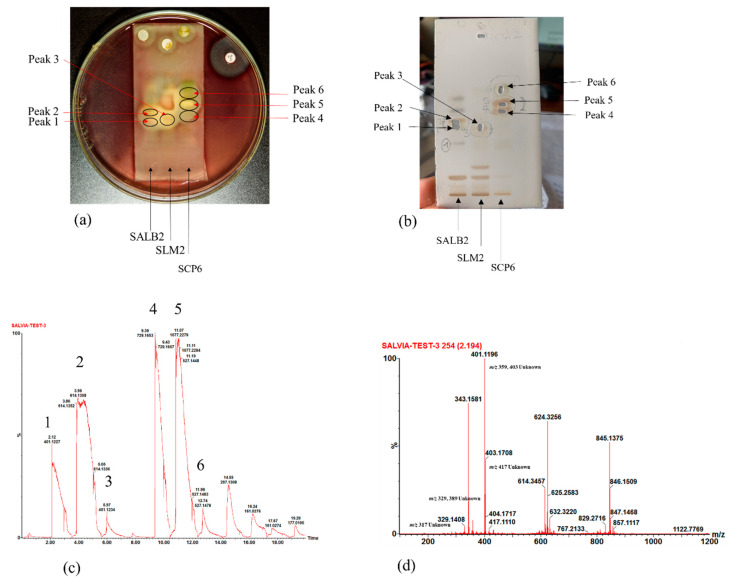
(**a**) Bioautograms of *S. africana-lutea* (SALB2), *S. lanceolata* (SLM2) and *S chamelaeagnea* (SCP6) extracts indicating the active sites indicated as Peak 1–6, corresponding to the peaks indicated in (**c**), with (**b**) the corresponding enlarged TLC plate showing areas where active constituents, Peak 1–6, were extracted using the TLC-sampler. (**c**) Chromatogram with Peaks 1–6 resulting from direct MS analysis of active bands extracted from the TLC plate and infused directly into the MS detector for ionisation. (**d**) Mass spectrum obtained, in this case for Peak 1, revealing molecular/fragment ions of compounds extracted from the active site on the bioautogram.

**Figure 4 antibiotics-11-00901-f004:**
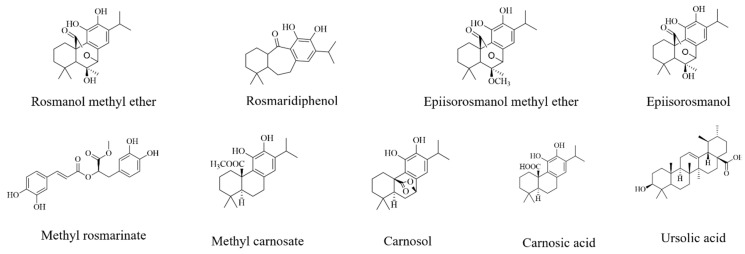
Chemical structures of compounds tentatively identified by TLC-MS analysis.

**Table 1 antibiotics-11-00901-t001:** Range and average (in brackets) minimum inhibitory concentrations (MICs) in mg/mL (*n* = 2) for crude *Salvia* extracts of each population (*n* = 5 for all populations except for Paarl where *n* = 6) tested against seven selected pathogens.

*S. africana-lutea* (*n* = 30)							
Population	*S. aureus* ATCC 25923	*B. cereus* ATCC 11778	*E. faecium* ATCC 27270	*B. subtilis* ATCC 6051	*A. baumannii* ATCC 19606	*P. aeruginosa* ATCC 27853	*E. coli* ATCC 8739
Atlantis	2.0–2.0 (2.0)	0.50–0.50 (0.50)	1.50–4.0 (3.3)	1.0–1.0 (1.0)	1.0–1.0 (1.0)	0.50–0.50 (0.50)	1.0–4.0 (3.1)
Silverstroomstrand	2.0–> 8.0	0.50–3.0 (1.4)	4.0–8.0 (5.6)	0.50–1.0 (1.4)	1.0–> 8.0	0.50–0.50 (0.50)	4.0–8.0 (4.8)
Betty’s bay	1.0–4.0 (2.7)	0.50–1.0 (0.70)	2.0–4.0 (3.0)	0.50–2.0 (1.1)	1.0–1.5 (1.1)	0.50–0.50 (0.50)	1.5–4.0 (3.3)
Blousteen	1.0–4.0 (2.5)	0.50–1.0 (0.80)	1.5–6.0 (3.5)	1.0–2.0 (1.4)	1.0–1.5 (1.3)	0.50–0.75 (0.55)	4.0–4.0 (4.0)
Rondeberg	2.0–4.0 (2.4)	0.25–0.50 (0.40)	1.0–2.0 (1.8)	1.0–2.0 (1.2)	0.50–1.0 (0.80)	0.50–0.75 (0.55)	1.0–2.0 (1.6)
Langebaan	1.0–3.0 (1.5)	0.25–0.38 (0.32)	0.50–2.0 (1.1)	1.0–2.0 (1.2)	0.25–1.0 (0.55)	0.50–0.50 (0.50)	1.0–2.0 (1.3)
Overall range of MICs (mg/mL)	1.0–> 8.0	0.25–3.0	0.50–8.0	0.50–2.0	0.25–> 8.0	0.50–0.75	1.0–8.0
Average of MICs ± SD (mg/mL)	2.9 ± 2.0	0.69 ± 0.53	3.0 ± 1.8	1.1 ± 0.4	1.2 ± 1.3	0.52 ± 0.08	3.0 ± 1.6
Positive control ciprofloxacin (µg/mL)	0.69	0.08	2.2	0.22	0.47	0.26	1.1
Number of active/more active samples (MIC ≤ 1.0 mg/mL)	6	28	5	23	23	27	6
Number of inactive/less active samples (MIC > 1.0 mg/mL)	24	2	25	7	7	3	24
***S. lanceolata* (*n* = 25)**							
Silverstroomstrand	2.0–> 8.0	1.0–3.0 (2.0)	4.0–> 8.0	0.50–0.50 (0.50)	1.0–2.0 (1.3)	0.25–0.50 (0.45)	4.0–6.0 (4.8)
Velddrif	1.0–3.0 (1.8)	0.50–1.0 (0.70)	2.0–5.3 (3.2)	0.50–0.50 (0.50)	0.50–1.0 (0.90)	0.13–0.50 (0.38)	2.0–6.0 (4.0)
Rondeberg	2.7–4.0 (3.7)	0.75–2.0 (1.1)	2.0–6.0 (4.4)	0.25–2.0 (0.95)	1.0–1.5 (1.1)	0.50–0.50 (0.50)	4.0–6.0 (4.4)
Yzerfontein	3.0–> 8.0	1.0–2.5 (1.9)	3.0–8.0 (4.2)	1.0–2.0 (1.2)	1.0–> 8.0	0.50–0.50 (0.50)	4.0–> 8.0
Mamre	1.0–4.0 (2.1)	0.38–1.0 (0.65)	0.75–4.0 (2.1)	1.0–1.0 (1.0)	0.25–1.5 (0.80)	0.25–0.50 (0.45)	1.0–4.0 (2.8)
Overall range of MIC values (mg/mL)	1.0–> 8.0	0.38–3.0	0.75–> 8.0	0.25–2.0	0.25–> 8.0	0.13–0.50	1.0–> 8.0
Average of MICs ± SD (mg/mL)	3.0 ± 1.8	1.3 ± 0.8	4.0 ± 2.0	0.83 ± 0.44	1.4 ± 1.4	0.46 ± 0.10	4.2 ± 1.5
Positive control ciprofloxacin (µg/mL)	0.69	0.080	2.2	0.22	0.47	0.26	1.1
Number of active/more active samples (MIC ≤ 1.0 mg/mL)	3	16	2	23	17	5	2
Number of inactive/less active samples (MIC > 1.0 mg/mL)	22	9	23	2	8	20	23
***S. chamelaeagnea* (*n* = 26)**							
Paarl	0.38–2.0 (0.98)	0.038–0.25 (0.16)	0.25–1.00 (0.57)	1.0–1.0 (1.0)	0.13–0.50 (0.25)	0.50–0.50 (0.50)	0.25–1.5 (0.69)
Simonsvlei	1.0–2.0 (1.2)	0.13–0.50 (0.20)	0.50–2.0 (1.0)	1.0–2.0 (1.6)	0.25–0.50 (0.40)	0.50–0.75 (0.55)	0.50–2.0 (1.0)
Du Toitskloof	0.75–4.0 (2.0)	0.13–0.50 (0.28)	0.25–2.0 (0.95)	1.0–1.0 (1.0)	0.25–1.0 (0.60)	0.50–0.50 (0.50)	0.50–4.0 (1.8)
Elandsberg	1.0–2.0 (1.7)	0.25–0.50 (0.38)	1.0–1.5 (1.2)	1.0–1.0 (1.0)	0.50–1.0 (0.60)	0.50–0.75 (0.55)	1.0–2.0 (1.3)
Riebeek Kasteel	1.0–1.0 (1.0)	0.13–0.25 (0.18)	0.50–1.0 (0.60)	1.0–2.0 (1.4)	0.25–0.50 (0.30)	0.50–0.75 (0.55)	0.50–1.0 (0.60)
Overall range of MIC values (mg/mL)	0.38–4.0	0.038–0.50	0.25–2.0	1.0–2.0	0.13–1.0	0.50–0.75	0.25–4.0
Average of MICs ± SD (mg/mL)	1.3 ± 0.8	0.23 ± 0.14	0.85 ± 0.48	1.2 ± 0.4	0.42 ± 0.24	0.53 ± 0.08	1.0 ± 0.8
Positive control ciprofloxacin (µg/mL)	0.69	0.080	2.2	0.22	0.47	0.26	1.1
Number of active/more active samples (MIC ≤ 1.0 mg/mL)	17	22	22	21	24	23	20
Number of inactive/less active samples (MIC > 1.0 mg/mL)	9	4	4	5	2	3	6

**Table 2 antibiotics-11-00901-t002:** Potentially bioactive compounds revealed through biochemometric analysis for active samples of *Salvia africana-lutea* towards seven selected pathogens.

Pathogen	OPLS-DA Model ^b^ Statistics				Compound ID	Rt (min)	[M-H]^−^ (*m*/*z*)	Molecular Formula	Compound Class	Correlates with HPTLC-MS
	A	R^2^X_P1_/R^2^X_O1_	R^2^X_cum_	Q^2^_cum_						
*S. aureus*	1 + 11	0.03/0.28	0.88	0.90	Methyl carnosate ^a^ Unknown	10.15 10.74	345, 346 389, 390, 329	C_20_H_26_O_4_ C_22_H_29_O_6_	Diterpenoid Diterpenoid	N/A
*B. cereus*	1 + 3	0.21/0.16	0.63	0.99	Methyl carnosate ^a^ Epiisorosmanol methyl ether ^a^	10.15 10.50	345, 346 359	C_20_H_26_O_4_ C_21_H_28_O_5_	Diterpenoid Diterpenoid	N/A
*E. faecium*	1 + 10	0.09/0.26	0.88	0.95	Dihydroxy-dimethoxyflavone derivative ^a^ Unknown	8.97 10.74	387 389, 390, 329	C_20_H_39_O_4_	Flavonoid	N/A
*B. subtilis*	1+ 7	0.04/0.27	0.82	0.84	Salvianolic acid E ^a^ Salvianolic acid B ^a^ Unknown	3.10 3.88 9.02	519 717, 519 417	C_36_H_30_O_16_ C_36_H_30_O_16_ C_21_H_37_O_8_	Caffeic acid derivative Caffeic acid derivative	N/A
*E. coli*	1 + 10	0.04/0.27	0.86	0.91	Unknown	10.74	389	C_22_H_29_O_6_	-	N/A
*A. baumannii*	1 + 7	0.04/0.27	0.80	0.89	Unknown Unknown	9.02 10.74	417 390, 329	C_21_H_37_O_8_ C_22_H_29_O_6_	-	Yes Yes
*P. aeruginosa*	1 + 8	0.02/0.29	0.84	0.89	Salvianolic acid B ^a^ Unknown	3.88 9.02	717, 519 417	C_36_H_30_O_16_ C_21_H_37_O_8_	Caffeic acid derivative	N/A

^a^ Tentative identification from literature; ^b^ Model significance and validity were confirmed by CV-ANOVA testing (*p* ≤ 0.05); N/A—Not applicable, TLC-DB not performed; A—number of predictive and orthogonal components; R^2^X_P1_—Variation of X-variables of predictive component; R^2^X_O1_—Variation of X-variables of orthogonal components; R^2^X_cum_—Variation of X-variables in terms of the cumulative value; Q^2^_cum_—Cumulative variation predicted by the model in specified component, according to cross-validation.

**Table 3 antibiotics-11-00901-t003:** Potentially bioactive compounds revealed through biochemometric analysis for active samples of *Salvia lanceolata* towards seven selected pathogens.

Pathogen	OPLS-DA Model ^b^ Statistics				Compound ID	Rt (min)	[M–H]^−^ (*m*/*z*)	Molecular Formula	Compound Class	Correlate with HPTLC-MS
	A	R^2^X_P1_/R^2^X_O1_	R^2^X_cum_	Q^2^_cum_						
*S. aureus*	1 + 8	0.06/0.17	0.70	0.93	Unknown Epiisorosmanol ^a^	6.79 7.80	331 345	C_20_H_26_O_5_	Diterpenoid	N/A
*B. cereus*	1 + 4	0.08/0.12	0.52	0.95	Unknown	8.72	331	C_21_H_31_O_3_		N/A
*E. faecium*	1 + 5	0.16/0.13	0.57	0.99	Epiisorosmanol ^a^ Unknown Unknown	7.80 10.36 10.43	345 317 331	C_20_H_26_O_5_ C_21_H_33_O_2_ C_21_H_31_O_3_	Diterpenoid	N/A
*B. subtilis*	1 + 3	0.04/0.19	0.40	0.53	Rosmarinic acid ^c^	3.66	359	C_18_H_15_O_8_	Caffeic acid derivative	N/A
*E. coli*	1 + 1	0.16/0.13	0.30	0.92	Epiisorosmanol ^a^ Unknown Unknown	7.80 10.36 10.43	345 317 331	C_20_H_26_O_5_ C_21_H_33_O_2_ C_21_H_31_O_3_	Diterpenoid	N/A
*A. baumannii*	1 + 7	0.04/0.17	0.67	0.96	Salvianolic acid E ^c^ Epiisorosmanol ^a^	3.10 7.80	717 345	C_36_H_30_O_16_ C_20_H_26_O_5_	Caffeic acid derivative Diterpenoid	No Yes
*P. aeruginosa*	1 + 5	0.05/0.18	0.57	0.92	Unknown Epiisorosmanol ^a^ Unknown	6.79 7.80 10.43	331 345 331	C_20_H_26_O_5_ C_21_H_31_O_3_	Diterpenoid	N/A

^a^ Tentative identification from literature; ^b^ Model significance and validity were confirmed by CV-ANOVA testing (*p* ≤ 0.05); ^c^ Identified by certified reference standard; N/A—Not applicable, TLC-DB not performed; A—number of predictive and orthogonal components; R^2^X_P1_—Variation of X-variables of predictive component; R^2^X_O1_—Variation of X-variables of orthogonal components; R^2^X_cum_—Variation of X-variables in terms of the cumulative value; Q^2^_cum_—Cumulative variation predicted by the model in specified component, according to cross validation.

**Table 4 antibiotics-11-00901-t004:** Potentially bioactive compounds revealed through biochemometric analysis for active samples of *Salvia chamelaeagnea* towards seven selected pathogens.

Pathogen	OPLS-DA Model ^b^ Statistics				Compound ID	Rt (min)	[M–H]^−^ (*m*/*z*)	Molecular Formula	Compound Class	Correlate with HPTLC-MS
	A	R^2^X_P1_/R^2^X_O1_	R^2^X_cum_	Q^2^_cum_						
*S. aureus*	1 + 9	0.21/0.27	0.89	0.94	Carnosol ^c^ Carnosic acid ^c^	10.00 11.39	329 331	C_20_H_26_O_4_ C_20_H_28_O_4_	Diterpenoid Diterpenoid	N/A
*B. cereus*	1 + 7	0.04/0.44	0.84	0.80	Carnosol ^c^ Carnosic acid ^c^ Unknown	9.99 11.39 12.19	329 331 317	C_20_H_26_O_4_ C_20_H_28_O_4_ C_21_H_33_O_2_	Diterpenoid Diterpenoid	N/A
*E. faecium*	1 + 12	0.03/0.44	0.91	0.88	Rosmanol methyl ether ^a^ Carnosol ^c^ Carnosic acid ^c^ Unknown	9.82 9.99 11.39 12.19	359 329 331 317	C_22_H_28_O_5_ C_20_H_26_O_4_ C_20_H_28_O_4_ C_21_H_33_O_2_	Diterpenoid Diterpenoid Diterpenoid	N/A
*B. subtilis*	1 + 7	0.04/0.43	0.84	0.82	Methyl carnosate ^a^ Unknown Unknown	10.15 10.74 12.19	345 389, 329 317	C_21_H_30_O_4_ C_22_H_29_O_6_ C_21_H_33_O_2_	Diterpenoid	N/A
*E. coli*	1 + 9	0.27/0.20	0.89	0.95	Carnosol ^c^ Carnosic acid ^c^	9.99 11.39	329 331	C_20_H_26_O_4_ C_20_H_28_O_4_	Diterpenoid Diterpenoid	N/A
*A. baumannii*	1 + 5	0.07/0.36	0.79	0.77	Epirosmanol ^a^ Rosmanol methyl ether ^a^ Carnosol ^c^ Carnosic acid ^c^	7.36 9.82 9.99 11.4	345 359 329, 330, 285 331	C_20_H_26_O_5_ C_22_H_28_O_5_ C_20_H_26_O_4_ C_20_H_28_O_4_	Diterpenoid Diterpenoid Diterpenoid Diterpenoid	Yes Yes Yes Yes
*P. aeruginosa*	1 + 8	0.02/0.44	0.86	0.78	Methyl carnosate ^a^ Unknown Unknown	10.15 10.74 12.19	345 389 317	C_21_H_30_O_4_ C_22_H_29_O_6_ C_21_H_33_O_2_	Diterpenoid	N/A

^a^ Tentative identification from literature; ^b^ Model significance and validity were confirmed by CV-ANOVA testing (*p* ≤ 0.05); ^c^ Identified by certified reference standard; N/A—Not applicable, TLC-DB not performed; A—number of predictive and orthogonal components; R^2^X_P1_—Variation of X-variables of predictive component; R^2^X_O1_—Variation of X-variables of orthogonal components; R^2^X_cum_—Variation of X-variables in terms of the cumulative value; Q^2^_cum_—Cumulative variation predicted by the model in specified component, according to cross validation.

**Table 5 antibiotics-11-00901-t005:** Molecular/fragment ions ([M-H]^−^ *m*/*z*) identified from active sites on bioautograms for each sample extract corresponding to peaks in Figure 3c.

Peak No.	Rt (min) (Figure 3c)	[M-H]^−^ *m*/*z*	Rt (min) from UPLC-MS	Possible Compound Identification (Compound Structures Given in Figure 4)	Correlate to Biochemometric Analysis
** *S. africana-lutea* ** **(SALB2)**					
1	2.12	417	9.07	Unknown	Yes
		359, 283, 329	9.81	Rosmanol methyl ether	No
		329	10.71	Unknown	Yes
		401		Unknown	No
		403, 343	11.50	Unknown	No
2	3.95	331	8.71	Unknown	No
		417	9.07	Unknown	Yes
		315	9.76	Rosmaridiphenol	No
		317	10.29/12.19/12.71	All three unknown	No
		331	10.49	Unknown	No
		315, 359	10.50	Epiisorosmanol methyl ether	No
		401		Unknown	No
		287	13.27	Unknown	No
** *S. lanceolata* ** **(SLM2)**					
3	5.97	403, 359	6.52	Unknown	No
		345	7.83	Epiisorosmanol	Yes
		315	9.80	Rosmaridiphenol	No
		359	10.50	Epiisorosmanol methyl ether	No
		329		Unknown	No
		347	8.55	Unknown	Yes
		375		Unknown	No
		383		Unknown	No
		401		Unknown	No
		433		Unknown	No
** *S. chamelaeagnea* ** **(SCP6)**					
4	9.39	343		Unknown	No
		373	4.57	Methyl rosmarinate	No
		287, 331	11.38	Carnosic acid	Yes
5	11.07	345	10.11	Methyl carnosate	No
		331	11.38	Carnosic acid	Yes
		343		Unknown	No
6	12.6	315	9.76	Rosmaridiphenol	No
		285, 329	9.98	Carnosol	Yes
		331, 287	11.38	Carnosic acid	Yes
		455	13.80	Ursolic acid	No

**Table 6 antibiotics-11-00901-t006:** Criteria used to separate the MIC values of pathogens that displayed moderate susceptibility to all extracts of the three species into classes for biochemometric analysis and identification of marker compounds.

Species	Pathogens	Class 1 (More Active) MICs	Class 2 (Less Active) MICs
*S. africana-lutea*	*P. aeruginosa*	≤0.50	>0.50
*S. lanceolata*	*P. aeruginosa*	≤0.38	>0.38
*S. chamelaeagnea*	*B. cereus* *A. baumannii* *P. aeruginosa*	≤0.38 ≤0.50 ≤0.50	>0.38 >0.50 >0.50

## Data Availability

Not applicable.

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
