# Peer review of "Antibacterial Screening, Biochemometric and Bioautographic Evaluation of the Non-Volatile Bioactive Components of Three Indigenous South African *Salvia* Species"

_antibiotics, 2022, doi:10.3390/antibiotics11070901_

Round 1

Reviewer 1 Report

The paper desribes the antibacterial features of active compounds isolated from local species from South Africa using biochemometric and other analytical methods. This investigation is appears relevant and was designed, performed and presented in the current manuscript in a convincing style and with all necessary informations that are necessary to understand the activity profiles. The biologically active molecules were MS-analyzed and could be identified as natural compound esters and acid. It would be nice to depict the molecular structures of these compounds also in the paper at a prominent place.

Author Response

Author response: Figure 4 has been added to the manuscript with the molecular structures for the compounds identified. See page 9 line 315 and 316 for introduction of Figure 4, and page 10 where the Figure is situated after Table 5.

Reviewer 2 Report

I have reviewed the manuscript entitled “Antibacterial screening, biochemometric and bioautographic evaluation of the non-volatile bioactive components of three indigenous South African Salvia species”.

This manuscript describes an in-depth investigation into the antibacterial activities of identify bioactive compounds in some Salvia species. The paper presents a useful application for exploration of a biochemometric approach, bioautography and direct MS analysis as useful tools for the rapid identification of bioactive compounds in plant extracts. It may be accepted for publication in “Antibiotics”.

Author Response

Author response: Thank you

Reviewer 3 Report

After reading the manuscript entitled "Antibacterial screening, biochemometric and bioautographic evaluation of the non-volatile bioactive components of three indigenous South African Salvia species", I consider it appropriate to be published in Antibiotics after minor corrections.

As a reviewer, I think the article is very valuable. The subject matter is original and important. The authors demonstrated the antibacterial activity of three plant extracts against both gram-positive and gram-negative bacteria. The authors showed that chromatographic analysis, microdilution test, analysis of biochemometric data, bioautography, and HPTLC-MS can be used to identify bioactive components present in medicinal plants.

The Introduction section presents background information, this part is concise and wellwritten. The methods present in the study are well described. The main results confirm the objectives proposed by the authors. References are correlated well with the text.

Minor corrections in the manuscript text must be performed to increase its quality.

page 2, lines 44-46 is: … including Staphylococcus…… (Gram-negative), should be: … including Gram-positive Staphylococcus aureus, Bacillus cereus, Enterococcus faecium, Bacillus subtilis, and Gram-negative Escherichia coli, Acinetobacter baumannii and Pseudomonas aeruginosa.…

page 2, line 51 is: … Staphylococcus aureus…, should be: … S. aureus

page 2, line 59 is: …have…, should be: … has…

page 4, lines 154-155 is:.. Activity was determined towards the pathogens that were highly susceptible toward the plant extracts, namely B. cereus and A. baumannii…– This sentence is not clear. Looking at the results in Table 1, it can be seen that P. aeruginosa was the most sensitive to S. africana-lutea (MIC=0.52 mg/mL) and S. lanceolata (MIC=0.46 mg/mL) extracts.

page 11, lines 369, 387 is: …an MIC…, should be: …a MIC…

page 11, line 382 is: …a main…, should be: …the main…

page 17, line 692 is: …towards…, should be: …to…

Author Response

All edits as requested have been actioned.

page 2, lines 44-46 is: … including Staphylococcus…… (Gram-negative), should be: … including Gram-positive Staphylococcus aureus, Bacillus cereus, Enterococcus faeciumBacillus subtilis, and Gram-negative Escherichia coli, Acinetobacter baumannii and Pseudomonas aeruginosa.…See lines 45-46

page 2, line 51 is: … Staphylococcus aureus…, should be: … S. aureus… The pathogen name is at the start of the sentence, and it is accepted journal policy to write it out in full, therefore it is kept as it is.

page 2, line 59 is: …have…, should be: … has…See line 60

page 4, lines 154-155 is:.. Activity was determined towards the pathogens that were highly susceptible toward the plant extracts, namely B. cereus and A. baumannii…– This sentence is not clear. Looking at the results in Table 1, it can be seen that P. aeruginosa was the most sensitive to S. africana-lutea (MIC=0.52 mg/mL) and S. lanceolata (MIC=0.46 mg/mL) extracts.

See page 4 line 155 – 157. The two pathogens were selected because individual extracts from S. chamelaeagnea displayed high activity towards the two pathogens with MICs as low as 0.038 and 0.13 mg/mL. That is now made clearer in the manuscript.

page 11, lines 369, 387 is: …an MIC…, should be: …a MIC…See line 376 and 394

page 11, line 382 is: …a main…, should be: …the main…See line 389

page 17, line 692 is: …towards…, should be: …to…See page 18 line 700